



# Understanding the variations and sources of CO, C$_2$H$_2$, C$_2$H$_6$, H$_2$CO and HCN columns based on three years of new ground-based FTIR measurements at Xianghe, China

Minqiang Zhou[1,2], Bavo Langerock[2], Pucai Wang[1], Corinne Vigouroux[2], Qichen Ni[1], Christian Hermans[2], Bart Dils[2], Nicolas Kumps[2], Weidong Nan[3], and Martine De Mazière[2]

[1]CNRC & LAGEO, Institute of Atmospheric Physics, Chinese Academy of Sciences, Beijing, China
[2]Royal Belgian Institute for Space Aeronomy (BIRA-IASB), Brussels, Belgium
[3]Xianghe Observatory of Whole Atmosphere, Institute of Atmospheric Physics, Chinese Academy of Sciences, Xianghe, China

**Correspondence:** Minqiang Zhou (minqiang.zhou@mail.iap.ac.cn; minqiang.zhou@aeronomie.be)

**Abstract.** Carbon monoxide (CO), acetylene (C$_2$H$_2$), ethane (C$_2$H$_6$), formaldehyde (H$_2$CO), and hydrogen cyanide (HCN) are important trace gases in the atmosphere. They are highly related to biomass burning, fossil fuel combustion and biogenic emissions globally, affecting air quality and climate change. However, the variations and correlations among these species are not well known in North China, due to limited measurements. In June 2018, we installed a new ground-based Fourier-transform

infrared (FTIR) spectrometer (Bruker IFS 125HR) recording mid-infrared high spectral resolution solar-absorption spectra at Xianghe (39.75 °N, 116.96 °E), China. In this study, we use the latest SFIT4 code, together with advanced a priori profile and spectroscopy, to retrieve these five species from the FTIR spectra measured between June 2018 and November 2021. The retrieval strategies, retrieval information and retrieval uncertainties are presented and discussed. For the first time, the time series, variations, and correlations of these five species are analyzed at a typical polluted site in North China. The seasonal

variations of C$_2$H$_2$ and C$_2$H$_6$ total columns show a maximum in winter-spring and a minimum in autumn, whereas the seasonal variations of H$_2$CO and HCN show a maximum in summer and a minimum in winter. Unlike the other four species, the FTIR measurements show that there is almost no seasonal variation in the CO column. The correlation coefficients (R) between the synoptic variations of CO and the other four species (C$_2$H$_2$, C$_2$H$_6$, H$_2$CO and HCN) are between 0.68 and 0.80, indicating that they are affected by common sources. Using the FLEXPART model backward simulations and satellite fire measurements, we

find that the variations of CO, C$_2$H$_2$, C$_2$H$_6$ and H$_2$CO columns are mainly dominated by the local anthropogenic emissions, while HCN column observed at Xianghe is a good tracer to identify fire emissions.

## 1 Introduction

Carbon monoxide (CO) is an important atmospheric trace gas, which affects air quality and the radiative balance of the Earth (IPCC, 2013). CO reacts with hydroxyl radicals (OH) and changes the atmospheric oxidizing capacity. CO is also a good trace

gas to study long-distance transport of fire and biomass burning emissions (Duflot et al., 2010; Zhou et al., 2018), as it has a relatively long lifetime of about 2 months (Khalil and Rasmussen, 1990). Atmospheric CO is mainly emitted from biomass





burning, fossil fuel combustion, and oxidation from methane ($CH_4$) and other biogenic non-methane hydrocarbons (NMHCs), and is removed mainly by the reaction with OH and partly by uptake by soil micro-organisms (Holloway et al., 2000). Ethane ($C_2H_6$) and acetylene ($C_2H_2$) are two major NMHCs. The sources of $C_2H_2$ and $C_2H_6$ are combustions from fossil fuel, biofuels and biomass burning, and the sink of $C_2H_2$ and $C_2H_6$ is the reaction with OH (Xiao et al., 2007, 2008). $C_2H_6$ is a strong source

of peroxyacetyl nitrate (PAN), a reservoir for nitrogen dioxide ($NO_2$). $NO_2$ normally has a short-lifetime, but in the form of PAN, it can be transported over long distances to a remote place leading to an important impact on the tropospheric ozone formation. The $C_2H_2$ oxidation by OH can form secondary organic aerosols, affecting the atmospheric chemistry (Volkamer et al., 2009). $C_2H_6$ has a similar lifetime as CO of about 2 months (Rudolph, 1995). $C_2H_2$ has a relatively short lifetime of 2 to 4 weeks (Xiao et al., 2007). Hydrogen cyanide (HCN) is a colourless and highly poisonous gas, less reactive than CO, with

a lifetime of 2-4 months. Atmospheric HCN's main sources are biomass burning, biogenic emissions and biofuels combustion, and it is mainly removed by the reaction with OH and ocean uptake (Li et al., 2000, 2003). Formaldehyde ($H_2CO$) is another important trace gas, mainly produced by methane and NMHCs oxidations in the atmosphere (Fortems-Cheiney et al., 2012). $H_2CO$ in the atmosphere reacts quickly with OH, $NO_3$, Cl, and Br, leading to a typical lifetime of a few hours (Anderson et al., 2017). $H_2CO$ is a major intermediate product in the degradation of isoprene in the atmosphere, and it strongly affects the

tropospheric ozone formation (Finlayson-Pitts and Jr., 1993). The main sources and sinks of global CO, $C_2H_2$, $C_2H_6$, $H_2CO$ and HCN are summarized in Table 1.

**Table 1.** Global sources and sinks of atmospheric CO, $C_2H_2$, $C_2H_6$, $H_2CO$ and HCN (Tg/year). The empty place means that the source or sink from that component is negligible.

| | CO | $C_2H_2$ | $C_2H_6$ | $H_2CO$ | HCN |
|---|---|---|---|---|---|
| Biomass burning | 748 | 1.6 | 2.4 | 0-10 | 1.4-2.9 |
| Fossil fuel/biofuels | 300 | 5.0 | 10.6 | 0-10 | 0.1-0.21 |
| Biogenic NMHC | 683 | | | 250 | 0-0.2 |
| Methane | 760 | | | 970 | |
| Total source | 2491 | 6.6 | 13 | ~1230 | 0.47-3.22 |
| Reaction with OH | 2261-2376 | ~6.6 | ~13 | ~1230 | 1.1-2.6 |
| Ocean uptake | | | | | 0.3 |
| Soil microorganisms | 115-230 | | | | |
| Total sink | ~2491 | ~6.6 | ~13 | ~1230 | 1.4-2.9 |
| Lifetime | 2 months | 2-4 weeks | 2 months | a few hours | 2-4 months |
| References | Holloway et al. (2000) | Xiao et al. (2007) | Xiao et al. (2008) | Fortems-Cheiney et al. (2012) | Li et al. (2003) |

Because of the common sources and sinks (Table 1), a number of studies have used the ratio of $C_2H_2$ to CO as a tracer of the age of air to investigate the relative importance of dilution and chemistry (Xiao et al., 2007; Parker et al., 2011). In addition,




the emission factors of biomass burning or forest fire were calculated and evaluated based on available aircraft observations (Goode et al., 2000; Xiao et al., 2007; Wetzel et al., 2021) and ground-based measurements (Zhao et al., 2002; Vigouroux et al., 2012; Lutsch et al., 2016). However, the variations and correlations of CO, $C_2H_2$, $C_2H_6$, $H_2CO$ and HCN in polluted area in North China are not well known due to limited measurements. It is expected that these species are strongly affected by

the anthropogenic emission there, but can we also capture the fire emissions?

In June 2018, we started measuring mid-infrared high spectral resolution solar-absorption spectra using a new ground-based Fourier-transform infrared (FTIR) spectrometer (Bruker IFS 125HR) at Xianghe (39.75 °N, 116.96 °E), China. Such FTIR measurements are compliant with the Network for Detection of Atmospheric Composition Change - InfraRed Working Group (NDACC-IRWG) (De Mazière et al., 2018) protocols. In contrast to most NDACC FTIR sites which are located at remote sites,

the Xianghe site is located in a highly-populated area, with strong anthropogenic emissions, among others from fossil fuel and biofuels combustion (Yang et al., 2020; Zhou et al., 2021). In this study, we present the first time series of FTIR retrievals of CO, $C_2H_2$, $C_2H_6$, $H_2CO$ and HCN at Xianghe, covering three years of measurements, and their use to investigate the variations of CO, $C_2H_2$, $C_2H_6$, $H_2CO$ and HCN columns on both seasonal and synoptic scales, as well as their correlations. Moreover, the FTIR measurements, atmospheric model and satellite measurements are used to understand the souses of CO, $C_2H_2$, $C_2H_6$,

$H_2CO$ and HCN columns in this region. Section 2 gives a brief introduction to the FTIR instrument and discusses the retrieval strategies, retrieval information and uncertainties. The time series, variations and correlations of CO, $C_2H_2$, $C_2H_6$, $H_2CO$ and HCN total columns are analysed in Section 3. the FLEXible PARTicle dispersion (FLEXPART) model is used to understand the sources of the observed airmasses at Xianghe. The VIIRS satellite observations are applied to locate the fire emission in the boreal forest. Based on the CO anthropogenic emissions and the ratios of $C_2H_2$ and $C_2H_6$ to CO derived from the FTIR

measurements, we derive the $C_2H_2$ and $C_2H_6$ emissions in North China and compare them to the inventories. Finally, the conclusions are drawn in Section 4.

## 2   FTIR measurements

### 2.1   Instrument

A Bruker IFS 125HR spectrometer was installed at Xianghe in 2016 and started recording solar absorption spectra in June 2018

(Zhou et al., 2020, 2021). The mid-infrared spectra covering the spectral range from 1800 to 5500 cm$^{-1}$ are recorded with an Indium antimonide (InSb) detector, cooled with liquid nitrogen. To increase the signal-to-noise ratio (SNR) of the spectrum used for specific target species, 6 NDACC-IRWG recommended optical filters (Blumenstock et al., 2021) are mounted in the wheel in front of the detector. Figure 1 shows three types of spectra (ch, hh, and nh) used in this study, and the main characters of these spectra are listed in Table 2.

To understand the optical status of the FTIR instrument, HBr cell spectra were recorded on 7 June 2018, 9 September 2018, 18 July 2019, 20 December 2019 and 13 September 2021. The instrument line shape (ILS) parameters: modulation efficiency (ME) and phase error (PE), are retrieved by LINEFIT14.5 algorithm (Hase et al., 1999). Figure 2 shows that the ME is relatively





**Table 2.** The characters of the 3 types of spectra observed at Xianghe used to retrieve CO, $C_2H_2$, $C_2H_6$, $H_2CO$ and HCN .

| spectrum type | ch | hh | nh |
|---|---|---|---|
| spectral range ($cm^{-1}$) | 2900-3600 | 2500-3200 | 1900-2700 |
| spectral resolution ($cm^{-1}$) | 0.0072 | 0.0051 | 0.0035 |
| Target species | $C_2H_2$, HCN | $H_2CO$, $C_2H_6$ | CO |
| SNR | 5000 | 1400 | 1600 |

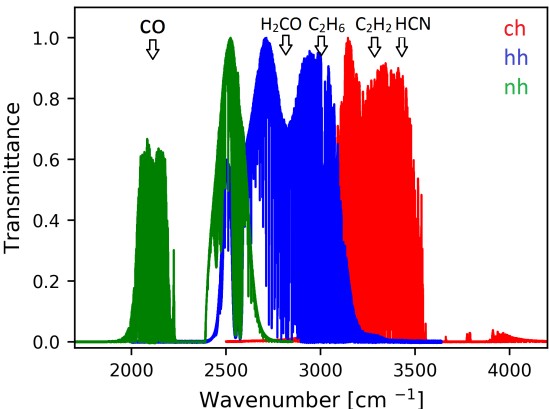

**Figure 1.** Normalized spectra (ch, hh, and nh) with three different filters used for CO, $C_2H_2$, $C_2H_6$, $H_2CO$ and HCN retrievals at Xianghe.

stable with about 0.86 at the maximum optical path difference (250 cm), and the PE is more variable with time. To reduce the retrieval uncertainty from the ILS, the LINEFIT outputs are used to represent the real ILS in the retrieval algorithm.

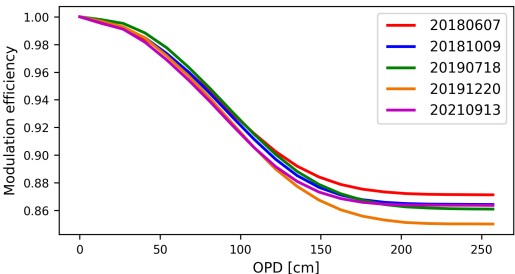
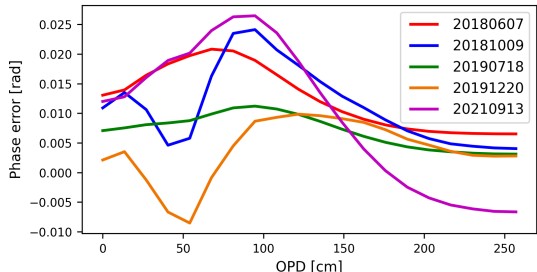

**Figure 2.** The retrieved modulation efficiency (left) and phase error (right) from the HBr cell measurements on 7 June 2018, 9 September 2018, 18 July 2019 and 13 September 2021 by LINEFIT14.5 algorithm.





## 2.2 Retrieval strategies

The SFIT4 retrieval code, updated from SFIT2 Pougatchev et al. (1995), is applied to retrieve CO, $C_2H_2$, $C_2H_6$, $H_2CO$ and HCN profiles from the FTIR spectra at Xianghe. The SFIT4 code is well-established and widely used in the Network for the Detection of Atmospheric Composition Change - Infrared Working Group (NDACC-IRWG). The retrieval algorithm includes
a line-by-line model $F(x, b)$ with the state vector $x$ (retrieved parameters) and non-retrieved parameters $b$ inputs to calculate the transmittance $Y$ at a given spectral range

$$Y = F(x, b) + \epsilon. \tag{1}$$

    Where $\epsilon$ denotes the error. There are 47 layers between the surface and the top of the atmosphere (100 km) included in the forward model. The retrieval windows for these species are listed in Table 3, and the absorption lines within each window
are shown in Figure 3. The retrieval windows are followed the NDACC-IRWG recommendation, and several previous FTIR studies (Zhao et al., 2002; Vigouroux et al., 2012; Viatte et al., 2014; Zhou et al., 2018; Vigouroux et al., 2018). Interfering species are retrieved along with the target species to reduce their impacts. The important interfering species are performed as a profile retrieval due to their strong absorption lines, and the other species are performed as a column retrieval (see Table 3). The atmospheric line list (ATM20), provided by Geoff Toon (JPL, NASA), is used for the CO, $C_2H_2$ and HCN retrievals,
the ATM18 is used for the $H_2CO$ retrieval, and the Pseudo line list, also provided by Geoff Toon (https://mark4sun.jpl.nasa.gov/pseudo.html; last access: 15 April 2022), based on the cross section made by Harrison et al. (2010), is used to retrieve $C_2H_6$. Note that the current NDACC-IRWG retrieval strategies for CO, $C_2H_2$ and HCN still use HITRAN08 spectroscopy (Rothman et al., 2009). We have tested both HITRAN08 and ATM20 for these three targets and their interfering species, and it is found that the fitting using ATM20 is improved, especially for CO, HCN and $H_2O$ (interfering). Therefore, we recommend
the ATM20 for these species. As mentioned above, to minimize the impact of the instrument, the LINEFIT results are used for the ILS inputs to represent the real instrument status. In addition, wavenumber shift, phase and slope are included in the state vector for all these five species. The solar intensity and solar shift are also retrieved for CO, $H_2CO$ and HCN because of relatively strong solar lines existed in their retrieval windows (see Figure 3). The vertical profiles of the temperature and water vapour are interpolated to the measurement time from the National Centers for Environmental Prediction (NCEP) 6-hourly
reanalysis data. Regarding $H_2CO$, we apply the harmonized retrieval strategy established by Vigouroux et al. (2018) for the NDACC-IRWG FTIR sites, where the average of the Whole Atmosphere Community Climate Model (WACCM) v6 monthly means between 1980 and 2020 is applied for the a priori profile. Regarding the other four species, the average of the WACCM v7 monthly means between 1980 and 2040 are applied for the a priori profiles of the target species and other interfering species except for $H_2O$, HDO, $H_2^{17}O$ and $H_2^{18}O$.
To solve this ill-posed problem (Eq.1) and to look for the optimal $x$ to make the simulated spectrum $F(x, b)$ close to the measured absorption spectrum $Y$, a cost function $J(x)$ is set up

$$J(x) = [y - F(x)]^T S_\epsilon^{-1}[y - F(x)] + [x - x_a]^T S_R[x - x_a]. \tag{2}$$



The cost function is determined from the measurement's error covariance $\mathbf{S}_\epsilon$, the a priori state $x_a$ and a regularisation matrix $\mathbf{S}_R$. The SNR is applied to calculate the measurement covariance $\mathbf{S}_\epsilon$, with the diagonal values equal to $1/\text{SNR}^2$, and non-diagonal values equal to zero. Regarding the regularization matrix $\mathbf{S}_R = \mathbf{S}_a^{-1}$, we use the optimal estimation method (OEM) and derive the covariance matrix from the WACCM monthly means as $\mathbf{S}_a$ for CO. For the other four species, we use the

Tikhonov $\mathbf{L}_1$ method ($\mathbf{S}_R = \alpha \mathbf{L}_1^T \mathbf{L}_1$). To determine the value of $\alpha$, we first use the WACCM monthly means to generate a covariance matrix, apply the retrieval using OEM, and then apply the retrieval using the Tikhonov method by tuning the $\alpha$ values to make the degree of freedom for the signal (DOFS) close to the OEM results (Steck, 2002). As a result, $\alpha$ values are set as 10, 100, 100, 30 for $C_2H_2$, $C_2H_6$, $H_2CO$ and HCN, respectively. After that, the retrieved state vector $\boldsymbol{x}_r$ can be calculated with the Levenberg-Marquardt iteration algorithm. Note that, the OEM or Tikhonov regularization is selected based

on the a priori knowledge. For CO, we have surface in situ measurements at Xianghe, which agree well with WACCM model simulations. The reason is probably that there are many surface in situ and aircraft profile measurements around the world (e.g. NOAA networks; HIPPO campaign). As the WACCM CO simulations perform well at Xianghe, we use the OEM method for CO and the a priori covariation matrix is derived from the WACCM model. However, for the remaining 4 species, we have no surface measurement at Xianghe and it is hard to evaluate the performance of the WACCM model simulations. Therefore, we

apply the Tikhonov L1 method for these targets, so that the retrievals are less affected by the a priori profiles.

**Table 3.** The retrieval window, interfering specie, spectroscopy, fitting parameters for these 5 species. Note that the isotopes of the $H_2O$ (HDO, $H_2^{17}O$ and $H_2^{18}O$) are treated as independent species in our retrieval.

| | CO | $C_2H_2$ | $C_2H_6$ | $H_2CO$ | HCN |
|---|---|---|---|---|---|
| Retrieval window ($cm^{-1}$) | 2057.7-2058.0 2069.56-2069.76 2157.5-2159.15 | 3250.5-3251.0 | 2976.66-2977.059 2983.2-2983.6 | 3268.14-3268.35 3331.4-3331.8 2778.15-2779.1 2780.65-2782.0 | 2763.42-2764.17 2765.65-2766.01 |
| Profile retrieval species | CO, $O_3$, $N_2O$ | $C_2H_2$, $H_2^{18}O$ | $C_2H_6$, $H_2O$ | $H_2CO$, $CH_4$, HDO | HCN, $N_2O$, $H_2^{17}O$ |
| Column retrieval species | OCS, $CO_2$, $H_2O$ | $H_2^{17}O$, $H_2O$ | $CH_3Cl$, $CH_4$, $H_2^{18}O$, $O_3$ | $H_2O$, $N_2O$, $O_3$ | $H_2O$, $O_3$, $CO_2$ |
| A priori profile | WACCM v7 | WACCM v7 | WACCM v7 | WACCM v6 | WACCM v7 |
| Spectroscopy | ATM20 | ATM20 | Pseudo | ATM18 | ATM20 |
| Retrieval parameters | wavenumber shift slope, phase solar shift & intensity | wavenumber shift slope, phase | wavenumber shift slope, phase | wavenumber shift slope, phase solar shift & intensity | wavenumber shift slope, phase solar shift & intensity |
| Regularization | OEM | Tik | Tik | Tik | Tik |
| DOFS | 2.4 | 1.5 | 1.5 | 1.4 | 2.2 |





**Figure 3.** The retrieval windows for CO (a,b,c), $C_2H_2$ (d), $C_2H_6$ (e,f), $H_2CO$ (g,h,i,j) and HCN (k,j). For each window, the bottom panel is the normalized transmittance from atmospheric species as well as the solar line, and the upper panel is the residual between the fitted and observed spectra. Note that, there is one de-weighting window (2157.775-2157.905 $cm^{-1}$) in CO band 3 and two de-weighting windows (2780.967-2780.993; 2781.42-2781.48 $cm^{-1}$) in $H_2CO$ band 4 , where we set the SNR as zero due to bad fittings.





## 2.3 Retrieval information

According to the optimal estimation method (Rodgers, 2000), the retrieved parameters can be written as

$$\boldsymbol{x}_r = \boldsymbol{x}_a + \mathbf{A}(\boldsymbol{x}_t - \boldsymbol{x}_a) + \epsilon, \tag{3}$$

where $\boldsymbol{x}_r$, $\boldsymbol{x}_a$ and $\boldsymbol{x}_t$ are the retrieved, a priori and true state vector, respectively; $\mathbf{A}$ is the averaging kernel (AVK), representing

the sensitivity of the retrieved parameters to the true statues; $\epsilon$ is the retrieval uncertainty, that will be discussed in Section 2.4. The a priori and retrieved profiles, together with the averaging kernels, are shown in Figure 4. For all these species, the retrieved profiles are mainly sensitive to the troposphere and lower stratosphere. There is a discontinuity found in the AVK of $C_2H_2$ and $C_2H_6$, because the mole fractions of $C_2H_2$ and $C_2H_6$ a priori profile (WACCM) become very low above 20 and 22 km, respectively. The pressure and temperature dependences of the absorption lines allow us to get some vertical information, but

the vertical resolution is limited. The trace of the averaging kernel is the DOFS, indicating the number of independent pieces of the information. The DOFS for these species are listed in Table 3. The DOFS of $C_2H_2$, $C_2H_6$, and $H_2CO$ are close to 1.5, and the DOFS of CO and HCN are close to 2.3, which indicates that we can derive 2 partial columns for CO and HCN at Xianghe. Based on the retrieved profile, we can obtain the total column

$$C_r = \boldsymbol{PC}_{air} \cdot \boldsymbol{x}_r = \boldsymbol{TC}_a + \boldsymbol{A}_c \cdot (\boldsymbol{PC}_t - \boldsymbol{PC}_a) + \epsilon_c, \tag{4}$$

where $\boldsymbol{PC}_{air}$ is the dry air partial column profile; $\boldsymbol{PC}_a$ and $\boldsymbol{PC}_t$ are the a priori and true partial columns of the target species; $\boldsymbol{A}_c$ is the column averaging kernel, representing the sensitivity of the retrieved total column to the true partial column profiles; $\epsilon_c$ is the retrieval uncertainty of the total column. If $\boldsymbol{A}_c = 1$ at one layer, the $C_r$ can well capture the deviation of the partial column from the a priori partial column ($\boldsymbol{PC}_t - \boldsymbol{PC}_a$) in that layer. If $\boldsymbol{A}_c < 1$ at one layer, the $C_r$ underestimates the deviation of the partial column from the a priori partial column in that layer, vice versa.

The a priori profiles and retrieved profiles (Figure 4) show that the mole fractions in the lower troposphere are larger than those in the higher altitude for all these species. Note that the CO mole fraction in the mesosphere is even larger than that in the lower troposphere due to the photolysis of carbon dioxide (Garcia et al., 2014), but it accounts for less than 1.0% of the total column. Based on the mean retrieved profile, the partial column between surface and 10 km accounts for 91.4%, 97.6%, 95.2%, 97.9% and 80.1% in the total column of CO, $C_2H_2$, $C_2H_6$, $H_2CO$ and HCN, respectively. The retrieved CO total column has

good sensitivity to the lower troposphere and stratosphere. The retrieved $C_2H_2$ and $C_2H_6$ total columns are mainly sensitive to the vertical range between the surface to 20 km, but they underestimate the deviation from the a priori partial column in the lower troposphere and overestimate the deviation from the a priori partial column in the upper troposphere and stratosphere. The retrieved $H_2CO$ total column has good sensitivity in the lower stratosphere, but overestimates the deviation from the a priori partial column above. The retrieved HCN total column is sensitive to the vertical range between the surface to about 40

km, where it underestimates the deviation from the a priori partial column in the lower troposphere and upper stratosphere, but overestimates the deviation from the a priori partial column in the upper troposphere and the lower stratosphere.





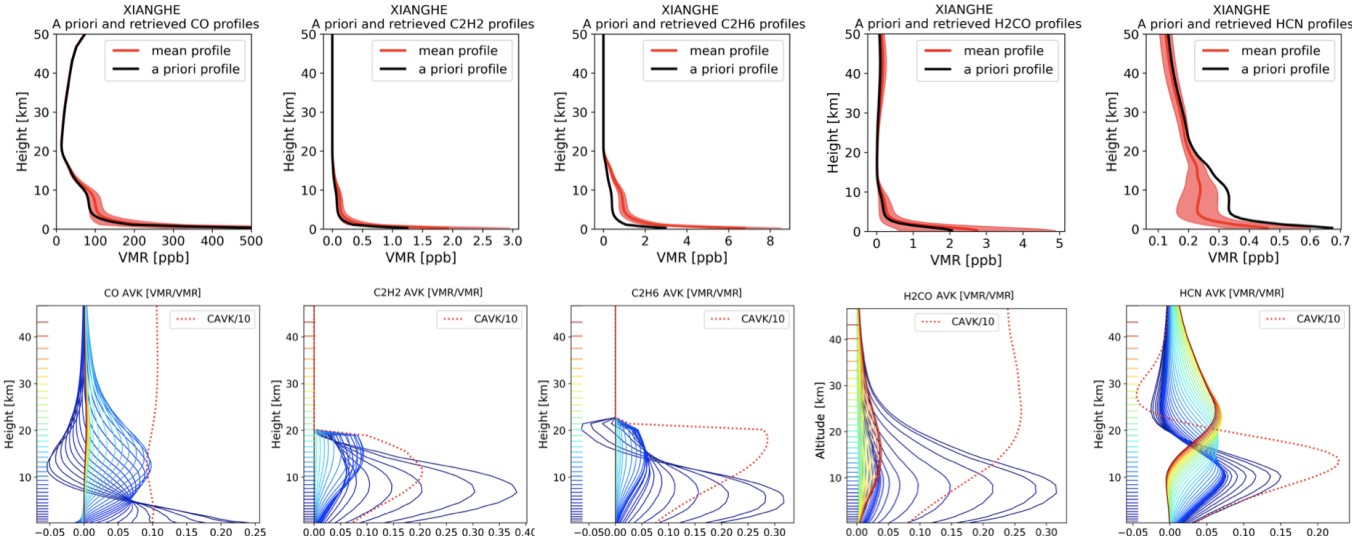

**Figure 4.** Upper panels: a priori profile (black) and retrieved profiles (red solid line: mean; red dashed area: standard deviation) for CO, $C_2H_2$, $C_2H_6$, $H_2CO$ and HCN. Lower panels: the averaging kernels and the column averaging kernels (divided by 10) for these species.

## 2.4 Retrieval uncertainty

The retrieval uncertainty on profile can be expanded as (Rodgers, 2000)

$$\epsilon_r = (\mathbf{A} - \mathbf{I})(\boldsymbol{x}_t - \boldsymbol{x}_a) + \mathbf{G}\mathbf{K}_b(\boldsymbol{b}_t - \boldsymbol{b}) + \mathbf{G}\epsilon_y, \tag{5}$$

where $\mathbf{A}$ is the averaging kernel of the state vector; $\boldsymbol{x}_t$ and $\boldsymbol{x}_a$ are the true and a priori state vector, respectively; $\mathbf{G}$ is the contribution function; $\boldsymbol{b}_t$ and $\boldsymbol{b}$ are the true and used model parameters, which affect the forward model but are not included in the state vector, such as spectroscopy, temperature, solar zenith angle (SZA) and zero offset (zshift); $\mathbf{K}_b = \frac{\partial \boldsymbol{F}(\boldsymbol{x}, \boldsymbol{b})}{\partial \boldsymbol{b}}$ is the weighting function of the model parameters, representing the sensitivity of the parameter $b$ to the forward model ($\boldsymbol{F}(\boldsymbol{x}, \boldsymbol{b})$); $\epsilon_y$ is the measurement noise.

The covariance of the retrieval uncertainty $\mathbf{S}_r$ can be written as

$$\mathbf{S}_r = (\mathbf{A} - \mathbf{I})\mathbf{S}_{\boldsymbol{a}}(\mathbf{A} - \mathbf{I})^T + \mathbf{G}\mathbf{K}_b\mathbf{S}_{\boldsymbol{b}}\mathbf{K}_b^T\mathbf{G}^T + \mathbf{G}\mathbf{S}_{\epsilon_y}\mathbf{G}^T. \tag{6}$$

On the right side of Eq. 6, the three items are the smoothing error, the model parameter error, and the measurement noise error, respectively. Here, we assume the forward model error is negligible. Note that the state vector ($\boldsymbol{x}$) not only contain the profile of the target species, but also the interfering species and other retrieved parameters (see Table 3). Therefore, we separate $(\mathbf{A} - \mathbf{I})(\boldsymbol{x}_t - \boldsymbol{x}_a)$ into three parts (Zhou et al., 2016): the smoothing error from the target gas uncertainty, the interfering species uncertainty, and other retrieved parameters uncertainty (slope, wavenumber shift, etc). For the model parameter error, we list the important parameters (spectroscopy, SZA, temperature and zshift) in Table 4. The SNR is applied to calculate the



measurement noise ($\epsilon_y = 1/\text{SNR}$). It is assumed that the measurement noise is purely random uncertainty. We need to set the uncertainties for other components based on a priori knowledge or other datasets, because $\boldsymbol{x}_t$ and $\boldsymbol{b}_t$ are not known. For the smoothing error estimation, we set 20% as the systematic uncertainties, and the random uncertainties are derived from the variations of WACCM model monthly means between 1980 and 2040. For the temperature profile, the systematic and random

uncertainties are calculated from the mean and standard deviation (std) of the differences between the NCEP and the European Centre for Medium-Range Weather Forecasts (ECMWF) ERA5 reanalysis data at Xianghe in the time period of 2016-2019. In general, the systematic uncertainty is about 1.0 K in the whole atmosphere and random uncertainty is about 2 K in the lower troposphere and about 1 K above. We estimate 0.1% and 0.5% for the systematic and random uncertainties of SZA, 1.0% for both systematic and random uncertainties of zshift. The uncertainty of the spectroscopy are set as 2.0 %, 5.0%, 10.0% and

10.0% for CO, $C_2H_2$, $H_2CO$ and HCN, respectively, based on the HITRAN2016 (Gordon et al., 2017). The uncertainty of the spectroscopy of $C_2H_6$ is set as 3%, according to the Pseudo line list (https://mark4sun.jpl.nasa.gov/pseudo.html). The total retrieval uncertainties of these species are also shown in Table 4. The systematic uncertainty is mainly from the spectroscopy, while the random uncertainty is mainly from smoothing error, SZA, temperature and zshift. The daily mean std of each species within $\pm 1$ hour around local noon with at least 2 measurements is calculated to represent the variability of the retrieval. In

general, the stds are close to the estimated random uncertainties for all species.

**Table 4.** The systematic and random (sys/ran) retrieval uncertainties for the total columns of five target species. The '-' means that the uncertainty is less than 0.1%. The std of each species within $\pm 1$ hour around local noon are shown in the bottom row to represent the variability of the retrieval.

| Uncertainty [%] | CO | $C_2H_2$ | $C_2H_6$ | $H_2CO$ | HCN |
|---|---|---|---|---|---|
| Smoothing-target species | 0.2/1.4 | 0.8/7.2 | 0.5/0.7 | 1.0/3.7 | 0.4/2.0 |
| Smoothing-interfering species | 0.1/0.1 | 0.3/0.5 | 0.1/0.7 | 0.3/0.7 | 0.1/0.6 |
| Smoothing-retrieved parameters | 0.1/0.1 | 0.7/0.7 | 0.1/0.1 | 0.1/0.1 | 0.1/0.1 |
| Measurement error | -/0.1 | -/2.1 | -/1.1 | -/2.0 | -/0.8 |
| Spectroscopy | 2.0/- | 5.5/- | 3.1/- | 11.2/- | 12.7/- |
| SZA | 0.1/0.6 | 0.1/0.4 | 0.3/1.2 | 0.2/1.1 | 0.4/2.0 |
| Temperature | 0.3/0.5 | 0.2/0.5 | 0.1/0.2 | 0.8/3.1 | 0.2/0.6 |
| Zshift | 0.2/0.2 | 2.2/2.2 | 1.4/1.4 | 6.0/6.0 | 1.2/1.2 |
| Total | 2.1/1.6 | 6.0/7.9 | 3.5/2.4 | 12.7/8.2 | 12.9/2.9 |
| std | 2.1 | 8.0 | 2.6 | 10.4 | 2.5 |



## 3 Results and Discussions

### 3.1 Time series and seasonality

The time series of the FTIR CO, $C_2H_2$, $C_2H_6$, $H_2CO$ and HCN total columns at Xianghe between June 2018 and November 2021 are shown in Figure 5. Due to the COVID-19 lockdown, the liquid nitrogen could not be delivered to the site. Therefore, there is a 3-months gap between 18 February and 23 May 2020. To better visualize the seasonal variation, the total columns are fitted by a periodic function $y(t)$

$$y(t) = A_0 + \sum_{k=1}^{3} (A_{2k-1} \cos(2k\pi t) + A_{2k} \sin(2k\pi t)), \tag{7}$$

where $A_0$ is the offset, and $A_1$ to $A_6$ are the periodic amplitudes, representing the seasonal variation.

– CO

The mean and std of CO total column are $2.86 \pm 0.87 \times 10^{19}$ molecules/cm$^2$. There is no clear seasonal variation, with a peak-to-peak amplitude of seasonal variation less than 5%. A similar seasonal variation of CO total column is observed by the TROPOspheric Monitoring Instrument (TROPOMI) onboard the S5P satellite measurements and Total Carbon Column Observing Network (TCCON) measurements at Xianghe (appendix A).

CO in the atmosphere is mainly emitted from biomass burning, biogenic and anthropogenic emissions, and removed by reaction with OH (Duncan et al., 2007). Significant seasonal variations of CO columns are observed at other FTIR sites (Zeng et al., 2012; Viatte et al., 2014; Té et al., 2016; Sun et al., 2020). For example, high values in spring and low values in autumn are observed by FTIR measurements in Paris, a mega-city in west Europe. Té et al. (2016) used the GEOS-Chem model to understand the seasonal variation of CO columns in Paris, and they found that the dominant factor is the anthropogenic emission, and more emissions are observed in spring. According to the anthropogenic emission datasets, such as the Regional Emission inventory in ASia (REAS) v3.2 (Kurokawa and Ohara, 2020) and the Emissions Database for Global Atmospheric Research (EDGAR) v5.0 (Crippa et al., 2018), the CO emissions in North China are high in winter and low in summer, with a relative difference of about 30%. Regarding the sink, more UV intensity in summer in North China (Hu et al., 2010) can generate a high OH level (Canty and Minschwaner, 2002). As a result, more OH in summer as compared to winter could also lead to a high CO column in winter. However, such a phase of the seasonal variation is hardly observed by the FTIR total columns. One possible reason is that the $CH_4$ mole faction is high in summer in North China, which is very different from other sites with a similar latitude (Karlsruhe, Germany and Lamont, USA) (Yang et al., 2020). $CH_4$ is also an important source of CO, and a high $CH_4$ level in summer at Xianghe will probably compensate for the impact of the lower anthropogenic CO emission and the stronger OH sink in summer. Another possible reason is that the Xianghe site is very close to the anthropogenic sources so the seasonal variation in OH has a weak effect on CO. To conclude, the small month-to-month variation in CO columns observed at Xianghe shows that the synoptic variation of CO columns is much stronger than the seasonal variation in North China.

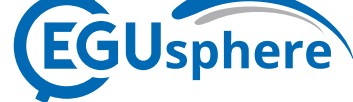

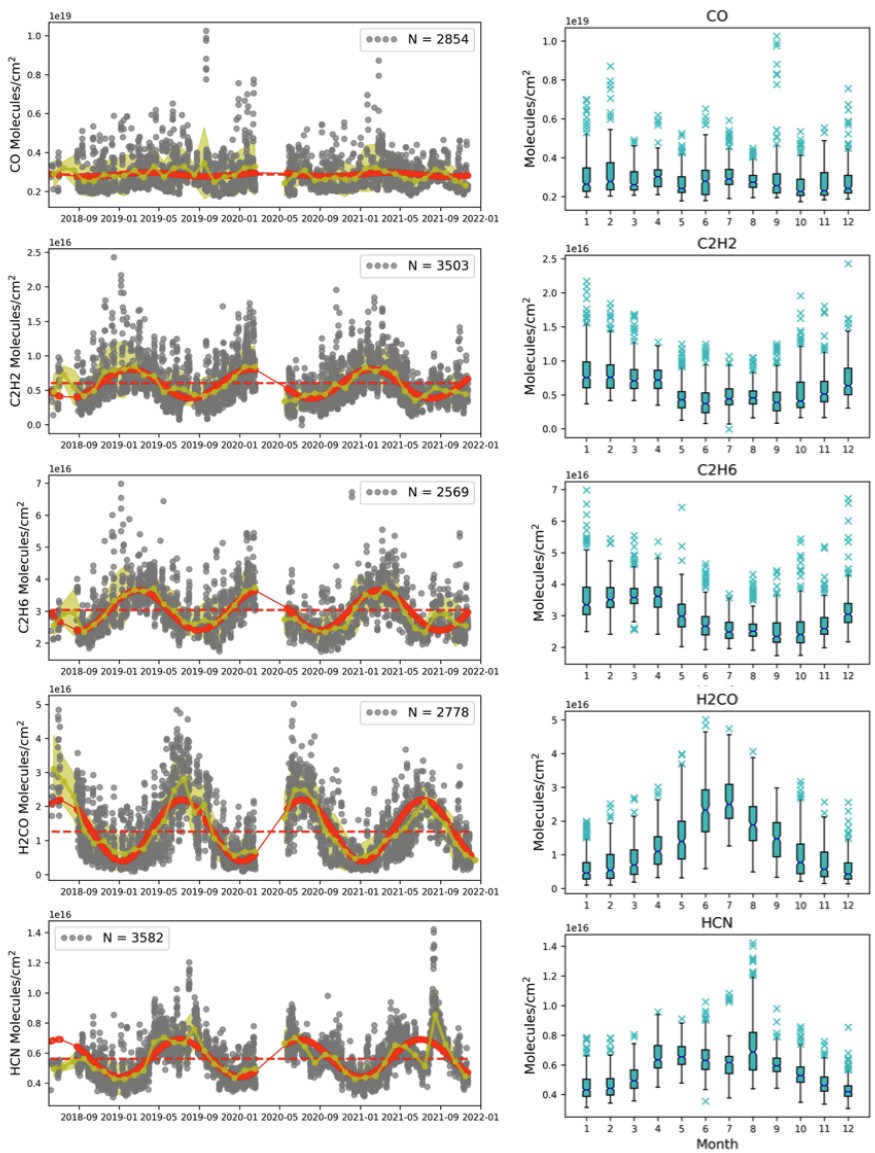

**Figure 5.** Left panels: time series of CO, C$_2$H$_2$, C$_2$H$_6$, H$_2$CO and HCN FTIR retrieved total columns between June 2018 and November 2021 at Xianghe. Grey dots are each individual retrievals with the total number indicated by N; the yellow dotted line is the monthly mean together with the yellow shaded area as the monthly standard deviation; the red dashed line is the offset $A_0$; the red solid line is the fitted time series $y(t)$. Right panels: the monthly box plot of the total columns. The bottom and upper boundaries of the box represent the 25% (Q1) and 75% (Q3) percentile of the data points around the median value, the errorbars extend no more than 1.5× IQR (IQR = Q3 - Q1) from the edges of the box, and the blue crosses are the outliers.





– $C_2H_2$ and $C_2H_6$

$C_2H_2$ and $C_2H_6$ belong to the Non-methane hydrocarbons (NMHCs). The mean and std of $C_2H_2$ and $C_2H_6$ total columns are $0.60\pm0.29 \times10^{16}$ molecules/cm$^2$ and $3.02\pm0.70 \times10^{16}$ molecules/cm$^2$, respectively. There is a clear seasonal variation in both species, with a high value in January - April and a low value in June - September. The peak-to-peak amplitudes of the fitted seasonal variations are $0.38 \times10^{16}$ molecules/cm$^2$ and $1.16 \times10^{16}$ molecules/cm$^2$ for $C_2H_2$ and $C_2H_6$, respectively, corresponding to 42% and 38% of their mean values. The major sources of $C_2H_2$ and $C_2H_6$ are biofuels and fossil fuel and the main sink is the reaction with OH. A slightly larger seasonal variation found in $C_2H_2$ as compared to $C_2H_6$ is consistent with the fact that $C_2H_2$ is removed more quickly by OH as compared to $C_2H_6$ (Xiao et al., 2007, 2008). According to the REAS v3.2 inventory, more $C_2H_2$ and $C_2H_6$ are emitted in winter as compared to summer, and the relative difference can reach 100%. The seasonal variations of $C_2H_2$ and $C_2H_6$ at Xianghe are in agreement with the measurements at two Japanese sites (Moshiri and Rikubetsu) (Zhao et al., 2002).

– $H_2CO$ and HCN

The mean and std of $H_2CO$ and HCN total column are $1.26\pm0.91 \times10^{16}$ molecules/cm$^2$ and $0.57\pm0.14 \times10^{16}$ molecules/cm$^2$, respectively. In contrast to the seasonal variation of $C_2H_2$ and $C_2H_6$, the seasonal variations of $H_2CO$ and HCN show a high value in summer and low value in winter. $H_2CO$ is mainly formed by the oxidation from methane and other hydrocarbons, which are emitted into the atmosphere by plants, animals, human activities and biomass burning. Shen et al. (2019) used the GEOS-Chem model to quantify the $H_2CO$ contributions in China, and they pointed out that $H_2CO$ are mainly generated from anthropogenic and biogenic sources. Based on the FTIR measurements at Xianghe, Ji et al. (2020) found that the $CH_4$ tropospheric column reaches to its maximum in summer, and the carbon-tracker model shows that there are large natural emissions (wetlands, soil, oceans, and insects/wild animals) in summer. HCN is a good tracer of biomass burning, and its lifetime is about 2-4 months (Li et al., 2000). The seasonal variation of HCN at Xianghe is similar to those observed at Kiruna, Toronto, Rikubetsu and Hefei (Sun et al., 2020) with two peaks around May and September. Sun et al. (2020) applied GEOS-Chem tagged CO simulation to understand that the peaks in May and September at Hefei are mainly from the biomass burning emissions at surrounding regions. FTIR measurements at Xianghe show that there is a low value of the HCN column in July 2021 but followed by a high peak in August 2021, which will be further discussed in Section 3.4.

Based on a bootstrap method (Gardiner et al., 2008), we also derive the relative annual changes of -2.2±2.0 ($2\sigma$) %/year, -1.6±2.0 %/year, -2.0±1.1 %/year, -6.7±4.0 %/year and 1.2±2.3 %/year for CO, $C_2H_2$, $C_2H_6$, $H_2CO$ and HCN total columns, respectively, between June 2018 and November 2021. HCN has a slightly positive trend, though it is not statistically significant. HCN is more affected by the biomass burning emission and has the longest lifetime among these species (Table 1). The other four species all show negative trends during the last 3.5 years, which is in agreement with the Chinese air pollution improvement (Zhang et al., 2019; Lu et al., 2020). However, due to the relatively short time coverage of about 3.5 years, the long-term trends of these species derived from the FTIR measurements are not further discussed in this study.



## 3.2 Correlation

According to EDGAR v5.0 and REAS v3.2 anthropogenic emission datasets, the spatial distributions of CO, $C_2H_2$, $C_2H_6$ and $H_2CO$ are very similar, with a large emission around Xianghe in North China. In this section, the correlations between CO and the other four species are investigated. CO is treated as a tracer gas here because the FTIR CO retrieval has the lowest random

uncertainty as compared to other species (Table 4). To reduce the impact of the atmospheric background and capture the day-to-day variation, the monthly median are removed from the total column ($\Delta$Gas = Gas - monthly median). High correlations are found between $\Delta$CO and other four species with the Pearson correlation coefficients (R) of 0.80, 0.70, 0.70, and 0.68 for $\Delta C_2H_2$, $\Delta C_2H_6$, $\Delta H_2CO$ and $\Delta$HCN, respectively. The intercepts of the four fitting lines are very close to zero, confirming that subtracting the monthly means can well remove the atmospheric background signal. The slope of the fitting represents the

enhanced ratio of the target gas column to the CO column.

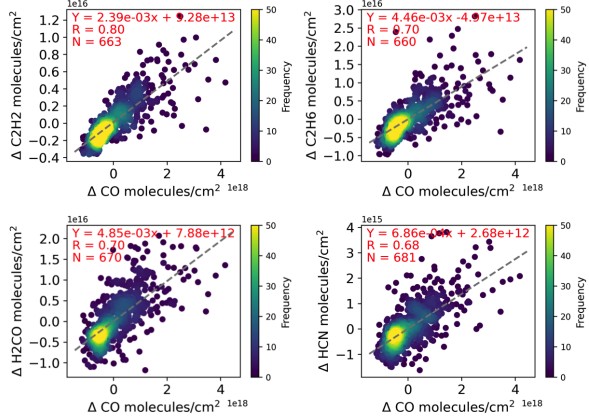

**Figure 6.** The correlation plot between $\Delta$CO (CO - monthly mean) and $\Delta C_2H_2$, $\Delta C_2H_6$, $\Delta H_2CO$ and $\Delta$HCN daily means. The dots are colored by the frequency. The grey dashed line is the linear fit; R is the Pearson correlation coefficient; N is the co-located number.

To check whether there is a seasonal dependence in their correlations, the correlation plots in four seasons are shown in Figure 7. It is noticed that we have fewer measurements in summer, because summer is the rainy season at Xianghe, and the FTIR measurements are only operated with a good weather condition. R values are relatively high in Spring, Autumn, and Winter. The relative low R values are found in summer for all species, which is due to the fact that less anthropogenic

emission, and more oxidation production in summer (Sun et al., 2020). Regarding the fitting slope, it is found that the slopes in summer are relatively less than those at the other three seasons for $C_2H_2$, $C_2H_6$, and HCN. The REAS v3.2 inventory provides CO, $C_2H_2$, and $C_2H_6$ monthly emissions, so that we calculate the monthly ratios of the $C_2H_2$ and $C_2H_6$ emissions to the CO emission in North China. The monthly emission ratios have a minimum in summer for both $C_2H_2$ and $C_2H_6$, which is generally in good agreement with our FTIR measurements. Different from the other three species, the lowest slope between

CO and $H_2CO$ is not observed in summer, but in winter. $H_2CO$ has a short lifetime and the formation of $H_2CO$ does not mainly come from the direct emissions but from the oxidation reaction.



**Figure 7.** Similar to Figure 6, but in four seasons.

### 3.3 Airmass source

In this section, the Lagrangian particle dispersion model FLEXPART v10.4 model is used to understand the airmass sources observed by the FTIR measurements at Xianghe. The backward running of the FLEXPART model releases the air particles at the Xianghe site and traces back to where the air mass comes from, providing source-receptor relationships. For more details about the FLEXPART v10.4 model, we refer to Pisso et al. (2019) and references therein. As the high mole fractions of these five species are located in the boundary height (see Figure 4), we release particles in the vertical range between the surface and 2 km. In addition, as the FTIR only does measurement during the daytime, we set the releasing temporal window as $\pm 1$ hour around local noon for all the FTIR measurement days between 1 June 2018 and 30 November 2021. The CO molecule is used as the trace gas. The main settings of FLEXPART are listed in Table 5. The model is driven by the National Centers for Environmental Prediction (NCEP) Climate Forecast System Version 2 (CFSv2) global dataset with a horizontal resolution of





$0.5° \times 0.5°$ and with 64 vertical levels from the surface to 0.266 hPa (Saha et al., 2014). We set the output layer between the surface and 500 m a.g.l., as it is related to the direct emission.

**Table 5.** The settings of FLEXPART v10.4 backward simulation used in this study.

| | |
|---|---|
| Input meteorological data | NCEP CFSv2 data |
| Tracer | CO |
| Release location horizontal | $\pm 0.1°$ latitude/longitude around Xianghe |
| Release location vertical | 0-2000 m a.g.l. |
| Release time | 11:00-13:00 local time for all FTIR measurement days |
| Number of days for backward running | 10 days |
| Number of particles for each release | 20000 |
| Output grid Horizontal | $0.5° \times 0.5°$ global |
| Output grid vertical | 0-500 m a.g.l. |

To better understand the airmass observed at Xianghe, the FTIR measurement days are classified into three categories (A, B, and C) according to different $\Delta CO$ levels. The background category A is selected as the following statistical method. Figure 8

shows the histogram of the $\Delta CO$ total column daily means at Xianghe. The blue dashed line is a Gaussian distribution ($\mu$=-0.4 and $\sigma$=0.3) fitted to the left side of the highest probability density (black dashed line). The green dashed line denotes $\mu + 2\sigma$. The $\Delta CO$ on the left side together with the $\Delta CO$ on the right side of the black line but less than $\mu + 2\sigma$ are generally following the fitted Gaussian distribution, so that the $\Delta CO$ value less than the green dashed line are selected as the category A. $\Delta CO$ distribution does not follow the Gaussian distribution when $\Delta CO$ is larger than the dashed green line. We use $4\sigma$ as the width

(purple dashed line: $\mu + 6\sigma$) to divide high $\Delta CO$ values into categories B (between green and purple dashed lines) and C (on the right side of the purple dashed line). The percentages of categories A, B and C are 75.3%, 21.7% and 3.0%, respectively. The mean and std of the CO vertical profiles of the three categories are also shown in Figure 8. As expected, a high variation in the CO mole fraction is observed in the lower troposphere. The CO mole fractions near the surface are about 0.4, 0.8 and 1.6 ppm in categories A, B and C, respectively.

Figure 9 shows the backward mean sensitivities for all the FTIR measurement days (701), categories A (528 days), B (152 days) and C (21 days). In general, the airmass observed in the 0-2 km vertical range above Xianghe is mainly coming from the west-north direction, with about 66% of the airmass sources in North China and Mongolia. The high emission response sensitivity (cyan circle) about 300 km around Xianghe accounts for 28% of the whole sensitivities. Regarding the category A, the major wind has a strong west-north direction. In addition, less airmass comes from the region surrounding Xianghe. The

airmass source inside the cyan circle takes up 26%, and the airmass sources in North China and Mongolia account for 60%. The rest 40% airmass are coming from Northeast China, Russia, Korea, Japan, middle Asia and Europe (not shown). For the



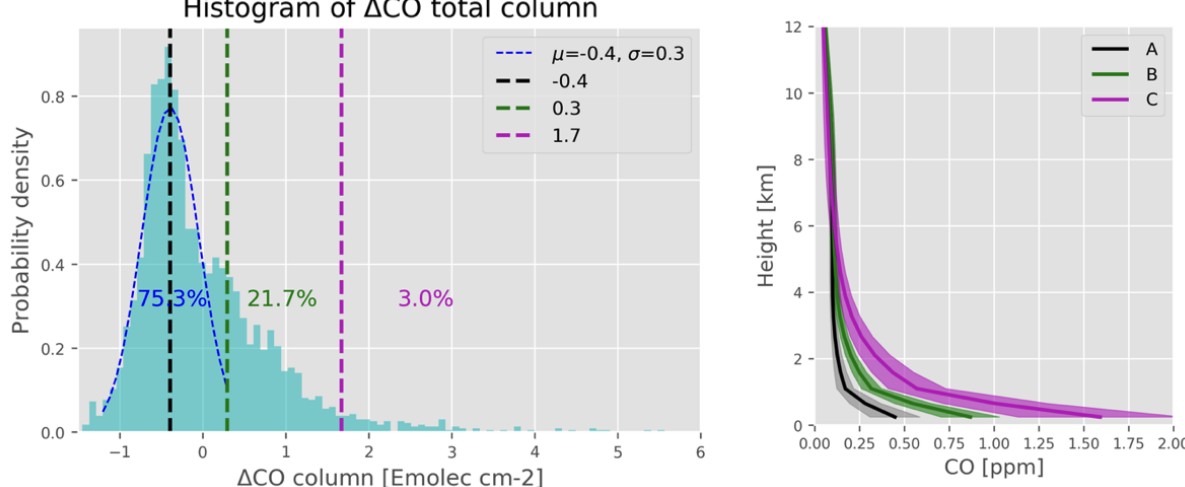

**Figure 8.** Left panel: the frequency density of the FTIR $\Delta$CO distribution. Based on the highest density, we apply a normal distribution fitting with ($\mu$=-0.4 and $\sigma$=0.3). The $\Delta$CO values less than $\mu + 2\sigma$, between $\mu + 2\sigma$ and $\mu + 6\sigma$, and larger than $\mu + 6\sigma$ are labelled as categories A (background), B, and C, respectively. The percentages of the categories A, B and C are also remarked in the figure. Right panel: the vertical profiles of CO mole fraction within the 3 categories.

category B, the high emission response sensitivity area (cyan and red lines) is generally like a circle around Xianghe, with 40% airmass coming from the cyan circle. The airmass sources in North China and Mongolia account for 75%. For the category C, the airmass sources in North China and Mongolia account for 89%. The high emission response sensitivity area (cyan and red lines) has a northeast to southeast direction. The area inside the red circle is of a high human activity level (Zhang et al., 2019),

corresponding to large anthropogenic emissions.

     In addition to the backward mean sensitivities, we also calculate the mean trajectory of all releasing particles of each model simulation. The mean and std of the airmass height, distance away from the releasing site, and the wind speed and direction of the categories A, B and C are shown in Figure 10. The wind speed is much larger in category A as compared to those in categories B and C, especially in the first two backward days. The major wind direction is northwest in category A, while the

major wind directions are the west and southwest during the first 2 days in categories B and C, respectively. After the first 2 backward days, the wind directions for the three categories are generally the same: wind coming from the northwest. Due to the different wind conditions during the first 2 backward days, the airmass between 0 and 2 km observed at Xianghe is coming from the airmass about 500 km, 190km and 130 km away from the site at 1 day ago, and about 1300 km, 520km and 410 km away from the site at 2 days ago for categories A, B and C, respectively. In terms of the airmass height, large differences are found

during the first 3 backward days where the airmass height is lower in category C and higher in category A. Our results show that the FTIR $\Delta$CO total column level is strongly impacted by the local meteorological condition. A high CO concentration is generally observed with a south-east wind direction, a wind speed below 10 km/h and a more stable atmospheric condition.





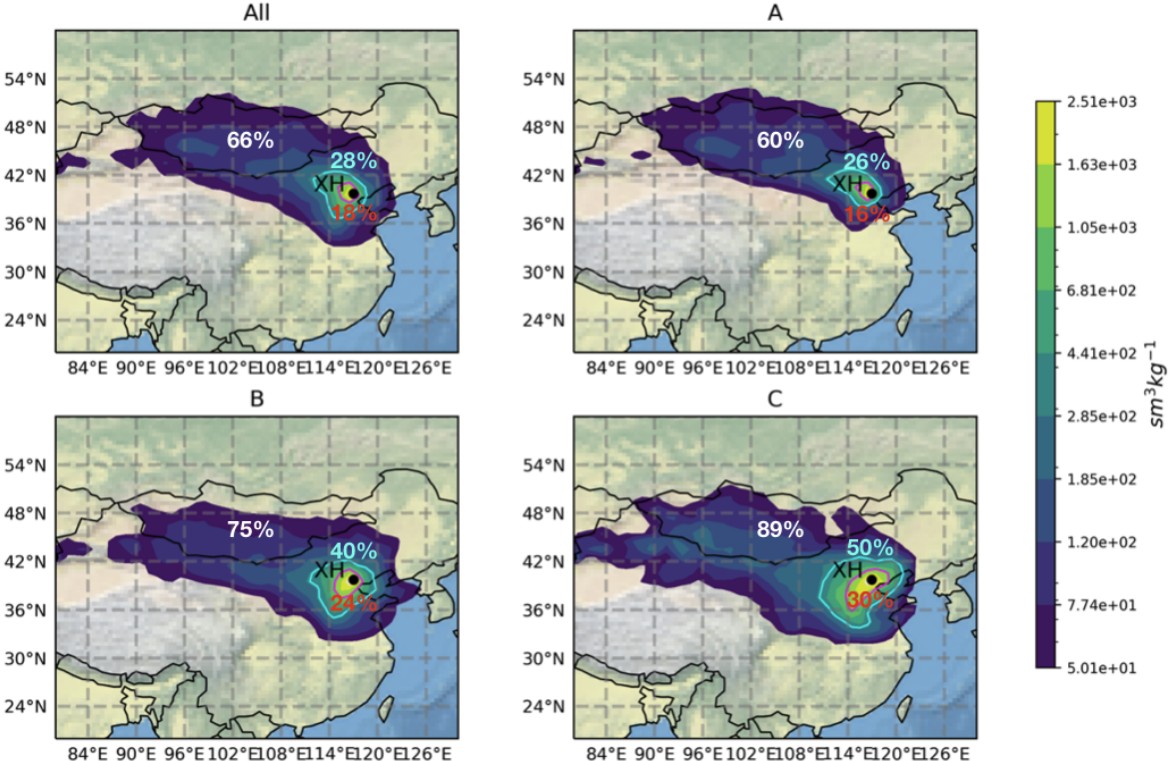

**Figure 9.** Spatial distributions of the airmass response sensitivity in 10 days' backward simulations at Xianghe using the FLEXPART v10.4 model for all FTIR measurement days (All) and categories A, B and C. Sensitivity is given in units of $sm^3kg^{-1}$. Xianghe site is remarked as the black dot. The values indicates the percentages of the emission response sensitivity larger than 50 (deep blue edge), 300 (cyan circle) and 1000 (red circle) $sm^3kg^{-1}$.

## 3.4 Fire emission observed by FTIR HCN measurements

The FTIR measurements observe a very strong HCN enhancement in August 2021, with the highest total column of $1.41 \times 10^{16}$ molecules/cm$^2$ on 11 August (Figures 5 and 11). Since HCN is a good trace of fire emission, such enhancement might be related to a fire activity captured by the FTIR measurements at Xianghe. In this section, the Visible Infrared Imaging

5 Radiometer Suite (VIIRS) data onboard the Suomi National Polar-orbiting Partnership (Suomi NPP) satellite Schroeder et al. (2014) are applied to understand the fire spatial distribution during this period. The VIIRS/SunomiNPP satellite measurements are capable to offer a complete coverage of the Earth during one day due to its large swath width of 3060 km. There are 22 spectral bands from 412 nm to 12 $\mu$m, among them 5 channels are imaging resolution bands (I-bands), which have a spatial resolution of 375m at the nadir Cao et al. (2014). In this study, we use the VIIRS 375m fire datasets, which are downloaded

10 from https://firms.modaps.eosdis.nasa.gov/. Note that we only select the fire pixels with confidence values equalling to normal



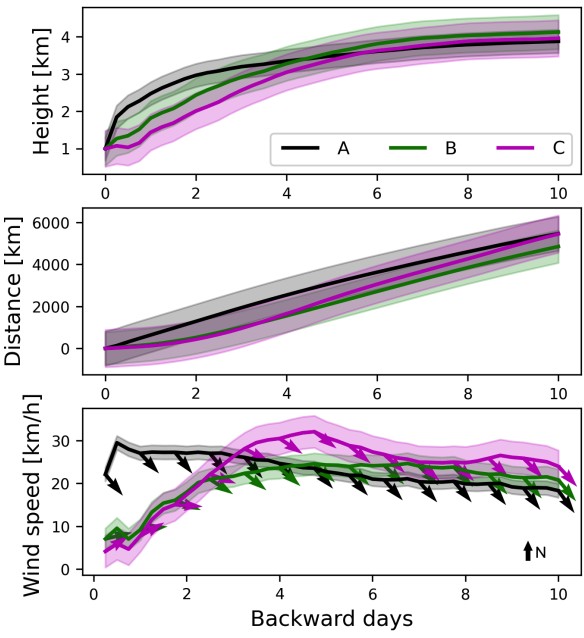

**Figure 10.** The mean (solid line) and std (shaded area) of the airmass height (upper panel), the distance to the Xianghe site (middle panel) and the wind direction and speed (bottom panel) of categories A, B and C simulated by the FLEXPART 10 days' backward running.

and high. The fire data are then binned into $0.5° \times 0.5°$ global grids, and the number of the individual fire pixels in each bin indicates the fire intensity.

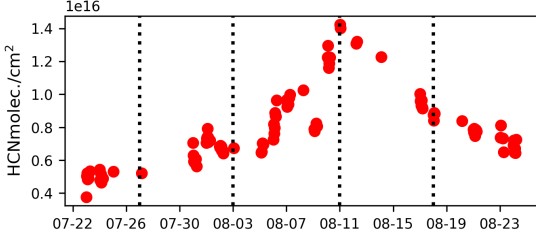

**Figure 11.** Time series of the FTIR HCN total column measurements at Xianghe between 22 July 2021 and 25 August 2021. Four days (27 July, 4 August, 11 August and 18 August) used to understand the airmass sources (see Figure 12) are remarked as the black dashed lines.

Figure 12 shows the airmass response sensitivity from the FLEXPART backward simulations on four HCN measurement days (27 July: low HCN column about two weeks before the peak; 3 August: 1 week before the peak; 11 August: the peak; 18 August: one weak after the peak), together with the fire counts observed by the VIIRS satellite. Similar to Section 3.3, the airmass is released at 0-2 km above Xianghe in the temporal window of $\pm 1$ hour around the local noon. Note that, in this section, we use the air as the trace gas instead of CO, and the output gird vertical is set to 0 - 5 km a.g.l. instead of 0-500 m




a.g.l.. All the individual fires within 1 and 10 days are summed up before each airmass releasing day. For example, the fire number map on 27 July 2021 is derived from all the fires between 18 July and 26 July 2021. Figure 12 shows that the area burned in the boreal forest of Russia (Siberia) between the end of July and August 2021 is related to the FTIR HCN peak. On 27 July, the airmass observed at Xianghe is coming from the southeast (Pacific ocean). Since the air in the Pacific ocean is clean,

the HCN total column on that day is very low of $0.52 \times 10^{16}$ molecules/cm$^2$. In the boreal forest region of Russia (Siberian and Far Eastern districts), we notice that there are several fire counts. On 3 August, the fire area in Siberia became large, and the airmass observed at Xianghe changed its main direction from southeast to northeast. However, only a small portion of the fire emission was transported and observed by the FTIR measurements at Xianghe, with the HCN total column of $0.69 \times 10^{16}$ molecules/cm$^2$. On 11 August, the fire area in Siberia remained large but slightly moved to the south, and the airmass observed

at Xianghe well captured the fire emission. As a result, the FTIR measurements observe the HCN total column peak. On 18 August, the boreal forest fire area got smaller and moved towards the north. The airmass observed at Xianghe is coming from the northeast, which only covers a part of the fire region. The HCN total column observed by the FTIR measurements decreases to $0.87 \times 10^{16}$ molecules/cm$^2$. Based on the satellite fire measurements and FLEXPART back simulations, we understand that the drop in HCN columns in July 2021 is due to the airmass coming from the Pacific ocean, and the peak of HCN columns in

August 2021 is due to the boreal forest fire emission in the Siberia.

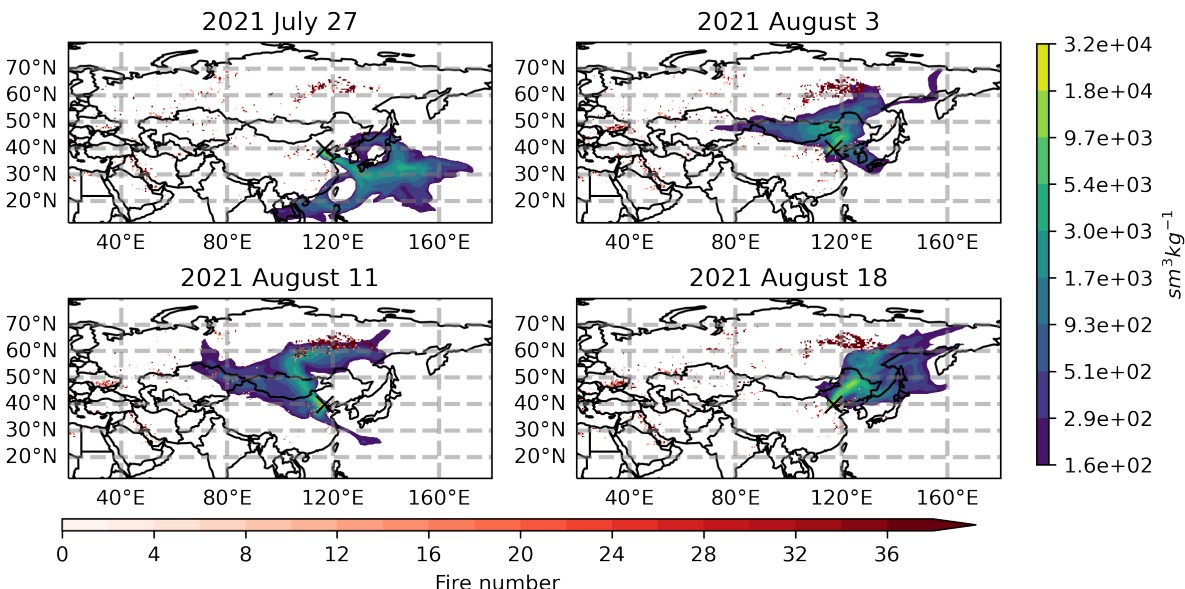

**Figure 12.** Spatial distributions of the airmass response sensitivity in 10 days' backward simulations at Xianghe using the FLEXPART v10.4 model on 27 July, 4 August, 11 August and 18 August 2021. The air is treated as trace gas and 20000 particles are released at 0-2 km above Xianghe in the temporal window of $\pm 1$ hour around the local noon. Xianghe site is remarked as the black cross. The fire counts are observed by the VIIRS/SunomiNPP satellite, and all the individual fires within 1 and 10 days before the airmass releasing day are combined and binned into $0.5° \times 0.5°$ global grids.




In fact, we can also observe enhancements in CO, $C_2H_2$ and $C_2H_6$ columns in August 2021 (Figure 5), although their amplitudes are not as significant as HCN. We do not observe any increase in $H_2CO$ columns at Xianghe on 11 August, which is due to that the fire emissions are less important for $H_2CO$ and the lifetime of $H_2CO$ is only a few hours (Table 1). The enhanced amplitudes in August 2021 are $1.65 \times 10^{18}$ molecules/$cm^2$ for CO, $5.57 \times 10^{15}$ molecules/$cm^2$ for $C_2H_2$, and $1.91$

$\times 10^{16}$ molecules/$cm^2$ for $C_2H_6$, respectively, which account for 39%, 42%, and 66% of their largest enhancements caused by the local anthropogenic emissions, respectively. To conclude, the enhancements of CO, $C_2H_2$ and $C_2H_6$ at Xianghe resulting from the fire emissions are less significant as compared to the local anthropogenic emissions, and the enhancements in CO, $C_2H_2$ and $C_2H_6$ caused by the fire emissions are less significant as compared to HCN. It indicates that HCN is a good trace gas for fire emissions in North China but not the other three species. The major emissions of CO, $C_2H_2$ and $C_2H_6$ at Xianghe

do not come from fire emissions but come from local anthropogenic emissions.

Apart from August 2021, can we also see the fire emission in other years? Figure 13 (a) shows the histogram distribution of the FTIR HCN column deviations (observations - seasonal variation). To identify the fire emission periods, we select all the HCN deviation larger than $2\sigma$ (on the right side of the orange vertical line). To further reduce the measurement uncertainty, we take the HCN-enhanced day with at least 3 individual measurements. As a result, we find another two periods (2019-04-17 to

2019-04-19 and 2019-07-31 to 2019-08-01) to investigate the fire emission (Figure 13 (b) and (c)). The enhancements of HCN in these two periods are less than that in August 2021. With the same method, the FLEXPART simulations and VIIRS satellite fire measurements are able to identify the fire emission sources (Figure 13 (d-i)). It is found that the enhancement of HCN in April 2019 is related to the fire emissions in Kazakhstan and Russia nearby, and the enhancement of HCN in July and August 2019 is related to the fire emissions in Siberia, Russia.

## 3.5 Emission estimation

The relationship of CO with NMHCs can provide useful information on the anthropogenic emissions (Wang et al., 2004; Liu et al., 2014). The high correlations among $\Delta CO$, $\Delta C_2H_2$ and $\Delta C_2H_6$ at Xianghe indicate that these species are commonly emitted from the surrounding sources. The FLEXPART model simulations show that the day-to-day variation of CO, $C_2H_2$ and $C_2H_6$ ($\Delta$gas) is insensitive to atmospheric transport, as the emissions are dispersed in a similar manner by the wind. Therefore,

the impact of transport cancels out in the ratio. The change in meteorological conditions is typically within 3-5 days, which is much shorter compared to the lifetime of CO, $C_2H_2$ and $C_2H_6$. Moreover, as Xianghe site is inside the anthropogenic emission region, the large enhancements of $\Delta CO$, $\Delta C_2H_2$, and $\Delta C_2H_6$ are directly coming from the local emissions, and the effect of OH sink is limited. Consequently, FTIR-observed ratios of $\Delta CO$ to $\Delta C_2H_2$ (or $\Delta C_2H_6$) at Xianghe (Figure 6) can directly link to emission ratios

$$\frac{E_g}{E_{CO}} = \frac{\Delta \text{FTIR}_g M_g}{\Delta \text{FTIR}_{CO} M_{CO}}, \tag{8}$$



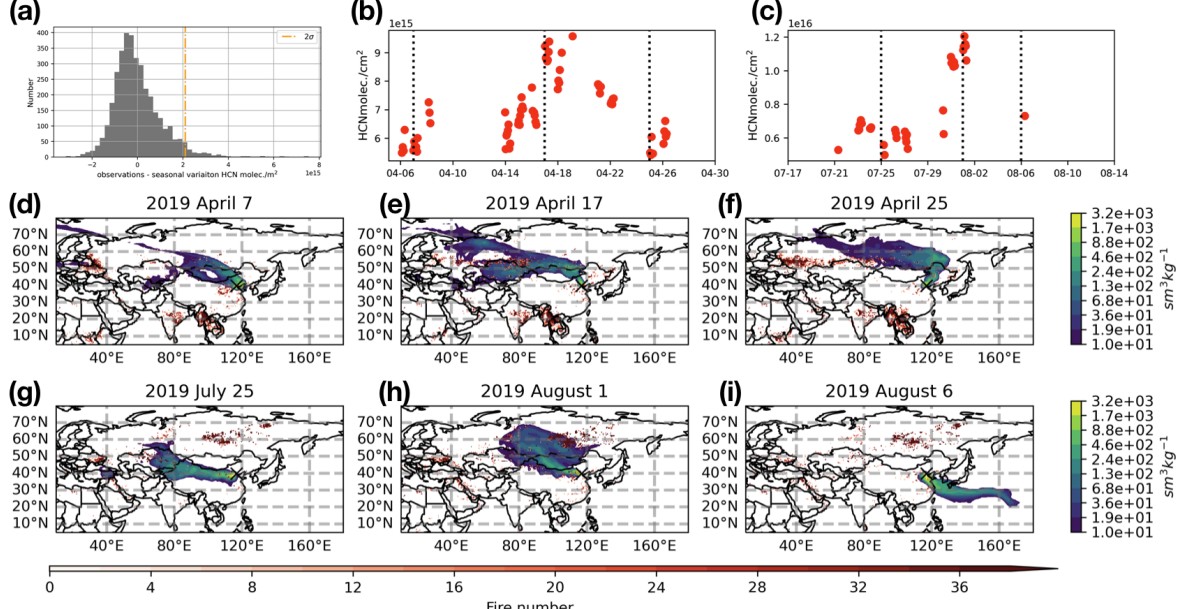

**Figure 13.** The frequency density of the FTIR HCN column variation (observations - seasonal variation) (a). The time series of the FTIR HCN total column measurements at Xianghe between 6 Apri 2019 and 30 April 2019 (b), and between 17 July 2019 and 14 August 2019 (c). For each period, three days (approximately one weak before the peak, peak and one weak after the peak) are used to understand the airmass sources for each HCN enhancement period are remarked as the black dashed lines. The spatial distributions of the airmass response sensitivity in 10 days' backward simulations at Xianghe using the FLEXPART v10.4 model (similar to Figure 10) on 7 April (d), 17 April (e), 25 April 2019 (f), and on 25 July (h), 1 August (i), 6 August 2019 (j).

where $E_{CO}$ is the emission of CO, $M_{CO}$ is the molecular mass of CO, and subscript $g$ represents $C_2H_2$ or $C_2H_6$. Note that $H_2CO$ and HCN are not discussed here, because $H_2CO$ is mainly generated from chemical reactions from $CH_4$ instead of the direct anthropogenic emission, and HCN is strongly affected by the fire emissions apart from anthropogenic emission.

Once we know the emission of CO, the emission of $C_2H_2$ or $C_2H_6$ then can be estimated based on Eq.8. According to the
5   FLEXPART backward simulation, we understand that the airmass observed by the FTIR measurements at Xianghe is mainly from North China and Mongolia (Figure 9). Here, we use the 66% of airmass response sensitivity for all FTIR measurement days (all in Figure 9) as the FTIR sensitive emission area (FSEA). Four global and regional inventories (EDGAR v5.0, REAS v3.2, Peking University emission inventory (PKU) (Zhong et al., 2017) and Multi-resolution Emission Inventory for China (MEIC) v1.3 (Li et al., 2017)) are applied to calculate the CO emission in the FSEA. Since these inventories have no data in
10  2018-2021, we use the latest available year for each dataset (EDGAR v5.0 and REAS v3.2 are 2015 annual means; PKU is the 2014 annual mean; MEIC v1.3 is the 2017 annual mean). Apart from the CO emission, the $C_2H_2$ and $C_2H_6$ emissions are also provided by the EDGAR v4.3.2 for 2012 annual mean and the REAS v3.2 for the 2015 annual mean.





Table 6 lists the CO emissions in the FSEA from the four inventories, and the corresponding $C_2H_2$ and $C_2H_6$ emissions calculated from the FTIR measurements using Eq.8 for these inventories. It is noticed that the CO emissions in the FSEA, particularly in North China, from these four inventories have a wide range from 36.8 to 80.6 $Mt/$year, which agrees well with the previous study (Zheng et al., 2018). As a result, the $C_2H_2$ and $C_2H_6$ anthropogenic emissions are estimated to be 0.08 -

0.17 $Mt/$year and 0.18 - 0.39 $Mt/$year, respectively. The $C_2H_2$ and $C_2H_6$ anthropogenic emissions from the EDGAR v4.3.2 are both 0.24 and 0.24 $Mt/$year, in which the $C_2H_2$ emission is larger than the result derived from the FTIR measurements, and the $C_2H_6$ emission is close to the result derived from the FTIR measurements. The $C_2H_2$ and $C_2H_6$ anthropogenic emissions from the REAS v3.2 are 0.11 and 0.12 $Mt/$year, respectively. In contrast to the EDGAR v4.3.2, the REAS v3.2 $C_2H_2$ emission is close to the result derived from the FTIR measurements, but the REAS v3.2 $C_2H_6$ emission is underestimated as compared

to the result derived from the FTIR measurements.

**Table 6.** The anthropogenic emissions of CO in the FSEA from the EDGAR v5.0 and REAS v3.2 in 2015, PKU in 2014 and MEIC v1.3 in 2017. $C_2H_2$ and $C_2H_6$ emissions are derived from the FTIR measurements at Xianghe together with the CO emissions.
[a] denotes the $C_2H_2$ and $C_2H_6$ emissions from the EDGAR v4.3.2 in 2012.
[b] denotes the $C_2H_2$ and $C_2H_6$ emissions from the REAS v3.2 in 2015.

| Emission Mt/year | EDGAR v5.0 (2015) | REAS v3.2 (2015) | PKU (2014) | MEIC v1.3 (2017) | $\Delta FTIR_g/\Delta FTIR_{CO} \times M_g/M_{CO}$ |
|---|---|---|---|---|---|
| CO | 36.8 | 70.3 | 80.6 | 50.0 | |
| $C_2H_2$ | $0.08\pm0.01$ $(0.24^a)$ | $0.15\pm0.01$ $(0.11^b)$ | $0.17\pm0.01$ | $0.11\pm0.01$ | $2.2\pm0.1\times10^{-3}$ |
| $C_2H_6$ | $0.18\pm0.01$ $(0.24^a)$ | $0.34\pm0.02$ $(0.12^b)$ | $0.39\pm0.02$ | $0.24\pm0.02$ | $4.8\pm0.3\times10^{-3}$ |

## 4    Conclusions

The variations and correlations of CO, $C_2H_2$, $C_2H_6$, $H_2CO$ and HCN columns in polluted region in North China are investigated based on the new FTIR measurements at Xianghe between June 2018 and November 2021. CO, $C_2H_2$, $C_2H_6$, $H_2CO$ and HCN columns are retrieved by the SFIT4 code followed by the NDACC-IRWG protocols. The retrieval strategies, retrieval

information and retrieval uncertainties of these species are well discussed. For all these species, the retrieved profile has good sensitivity in the troposphere. According to the DOFS, there are more than 2 pieces of independent information in CO and HCN, and mainly the column information in $C_2H_2$, $C_2H_6$, and $H_2CO$. In this study, we mainly focus on the total columns, and the systematic and random uncertainties of CO, $C_2H_2$, $C_2H_6$, $H_2CO$ and HCN columns are 2.1±1.6%, 6.0±7.9%, 3.5±2.4%, 6.0±6.0%, 12.9±2.9%, respectively.

The mean and std are $2.86\pm0.87\times10^{19}$ molecules/cm$^2$, $0.60\pm0.29\times10^{16}$ molecules/cm$^2$, $3.02\pm0.70\times10^{16}$ molecules/cm$^2$ $1.26\pm0.92\times10^{16}$ molecules/cm$^2$ and $0.57\pm0.14\times10^{16}$ molecules/cm$^2$ of CO, $C_2H_2$, $C_2H_6$, $H_2CO$ and HCN columns, respectively. The seasonal variations of $C_2H_2$ and $C_2H_6$ total columns show a maximum in winter-spring and a minimum in



autumn, and the seasonal variations of $H_2CO$ and HCN show a maximum in summer and a minimum in winter. We find that there is no clear seasonal variation in the CO total column with a month-to-month variation of less than 5% (consistent with TROPOMI satellite measurements and TCCON measurements). The weak seasonal variation of the CO column at Xianghe is very different from other FTIR sites, such as Paris and Hefei, which is probably to a large $CH_4$ value in summer at Xianghe

(Ji et al., 2020; Yang et al., 2020), and a short distance between the Xianghe FTIR site and the local anthropogenic sources. A more dedicated investigation, involving atmospheric transport chemical models (e.g. GEOS-Chem or WRF-Chem), is planned in the near future to quantify these contributions to the CO column seasonal variation at Xianghe.

The FLEXPART model is applied to understand the airmass sources observed by the FTIR measurements at Xianghe. It is found that the high values of these species are mainly coming from the local anthropogenic emissions in North China. With a

strong wind speed (typically larger than 20 km/h at 1 km a.g.l) and northwest wind direction, the FTIR measurements show lower values. Together with the VIIRS satellite fire measurements, we found that the fire emissions in boreal forest (Kazakhstan and Russia) can transport to Xianghe and be well captured by our FTIR measurements, leading to a strong enhancement in FTIR HCN columns. The FTIR HCN column is a good trace gas for fire emissions in North China but not the other four species. The major emissions of CO, $C_2H_2$ and $C_2H_6$ at Xianghe do not come from fire emissions but come from local anthropogenic

emissions. The high correlations between CO, $C_2H_2$ and $C_2H_6$, with R between 0.70 and 0.80, indicate that they are affected by the common sources, mainly from the local anthropogenic emissions. We use the slopes observed by the FTIR measurements to estimate their emissions. Based on four global and regional CO emission inventories (EDGAR v5.0, REAS v3.2, PKU, and MEIC v1.3), the CO emission in the FSEA (North China and Mongolia) is between 36.8 and 80.6 $Mt/\text{year}$. The $C_2H_2$ and $C_2H_6$ anthropogenic emissions in the FSEA are estimated to be 0.08 - 0.17 $Mt/\text{year}$ and 0.18 - 0.39 $Mt/\text{year}$, respectively.

The CO, $C_2H_2$, $C_2H_6$, $H_2CO$ and HCN FTIR column measurements at Xianghe are helpful to investigate the air quality in North China. Although the time coverage now is still too short to understand their long-term trend, the ongoing FTIR measurements at Xianghe can help us to study the Chinese air pollution improvement, long-distance fire emissions transportation, satellite calibration and validation, and model verifications.

*Data availability.* The EDGAR, REAS, PKU and MEIC emission inventories are publicly available at their corresponding websites. The

VIIRS fire data are publicly available via NASA datasets https://firms.modaps.eosdis.nasa.gov/. The FTIR measurements are available upon request to the authors.

*Competing interests.* The authors declare that they have no conflict of interest.

*Acknowledgements.* This study is supported by the National Natural Science Foundation of China (41975035) and the National key research and development program (2021YFB3901000). The authors would like to thank Qun Chen and Qing Yao for operating the FTIR measure-

ments at Xianghe, and the NDACC-IRWG community, especially James Hannigan (NCAR) and Mathias Palm (University of Bremen), for providing the SFIT4 v1.0 code and WACCM model simulations.

*Author contributions.* MZ wrote the manuscript and designed the experiment. MZ, BL, PW, CV, MDM discussed the conceptualization. CH, NK, WN collected the FTIR measurements at Xianghe. All the authors read and commented on the manuscript.

## Appendix A: TCCON and TROPOMI CO measurements at Xianghe

The FTIR NDACC-type measurements at Xianghe show that the seasonal variation of CO total column is very weak, which is very different from other FTIR sites. Since the TCCON measurements are also operated at Xianghe using the same Bruker IFS 125HR spectromete (Yang et al., 2020). Here, we check the dry-air column-average mole fraction of CO (XCO) time series and

5    seasonal variation from the TCCON measurements (GGG2020) at Xianghe. Moreover, we also select the TROPOMI overpass offline CO column data (Borsdorff et al., 2019) within 50 km around Xianghe site between June 2018 and November 2021 (https://scihub.copernicus.eu/). The spatial resolution of the TROPOMI measurement is $7.0 \times 7.0$ km$^2$ before 6 August 2019 and of $7.0 \times 5.5$ km$^2$ afterwards. Figure A1 shows the time series and seasonal variation of the CO column observed by the TCCON and TROPOMI satellite at Xianghe. Both TCCON and TROPOMI measurements show that the seasonal variation

10    of CO column at Xianghe is hardly observed. The amplitudes of the fitted seasonal variation derived from the TCCON and TROPOMI measurements at Xianghe are within 6%, which is similar to the ground-based FTIR NDACC-type measurements.

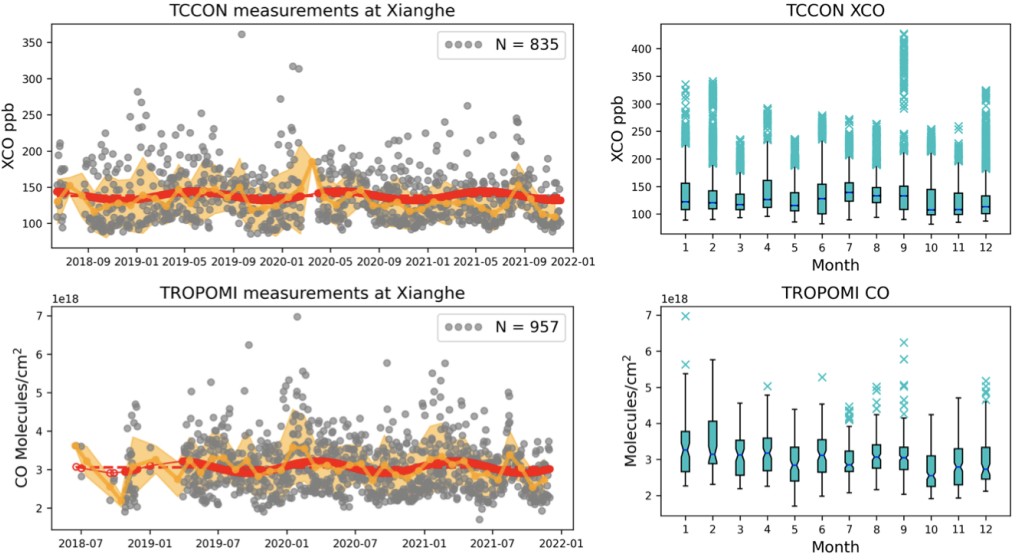

**Figure A1.** Left panels: time series of XCO and CO columns observed by TCCON at Xianghe (top) and TROPOMI overpass within 50 km around Xianghe (bottom) between June 2018 and November 2021. Grey dots are daily means with the total number indicated by N; the orange dotted line is the monthly mean together with the yellow shaded area as the monthly standard deviation; the red dashed line is the offset $A_0$; the red solid line is the fitted time series $y(t)$. Right panels: the monthly box plot of the CO columns in each month. The bottom and upper boundaries of the box represent the 25% (Q1) and 75% (Q3) percentile of the data points around the median value, the errorbars extend no more than $1.5 \times$ IQR (IQR = Q3 - Q1) from the edges of the box, and the blue crosses are the outliers.



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
