# Peer review of "Understanding the variations and sources of CO, $C_2H_2$ , $C_2H_6$ , $H_2CO$ and HCN columns based on three years of new ground-based FTIR measurements at Xianghe, China"

_EGUsphere, 2022_

## Author Comment (AC1)

*Black: referee's comments green: authors' answers*
*First of all, we want to thank the two referees for taking the time to review our paper and for their helpful comments. For the details, please look into the paper with keeping track of changes.*

Referee #1

This paper shows the Carbon monoxide (CO), acetylene (C2H2), ethane (C2H6), formaldehyde (H2CO), and hydrogen cyanide (HCN) column retrievals derived from the ground-based FTIR measurements at Xianghe, China. Such new ground-based FTIR datasets, following NDACC protocol with high precision and accuracy, are very important to understand these atmospheric components in this region. Until now, the variations and correlations among these species are not well known in North China, since limited or even no column measurements are available. The measurement and retrieval techniques of the ground-based FTIR dataset are nicely presented and well discussed. The seasonal variations of C2H2, C2H6, H2CO, and HCN are similar to other places (previous studies), while there is almost no seasonal variation of CO at Xianghe, which is different from other places. The paper shows that this weak seasonal variation of the CO column is also observed by co-located TROPOMI satellite and ground-based TCCON measurements. The HCN columns observed at Xianghe are also applied to identify the fire emission in Russia and Kazakhstan. In general, the paper is well-written, and the results are summarized well with novel scientific founds. Therefore, I would like to recommend it to AMT after addressing the following minor comments.

Minor comments:

P5,Eq1, change the '.' to ',' 'Where' to 'where'

Done

P10 line 13-14 "The daily mean std of each species within ±1 hour around local noon with at least 2 measurements is calculated to represent the variability of the retrieval. " I guess I understand what the authors did, but it is confusing from this sentence. Please rewrite it.

Done

P11. Eq 7, please add an uncertainty component in the formula.

Done

P13. line 27 – 34. As the uncertainty of the trend is very large due to limited time coverage, and the trend of these species is not the key point of this paper. I would suggest removing this paragraph.

Thanks for the suggestion. The discussion about the long-term trend is removed now.

Please check the reference carefully, as some references are not correctly formatted. For example, P30 line 20. This reference has two times of doi.

Done

---

## Author Comment (AC2)

*Black: referee's comments green: authors' answers*
*First of all, we want to thank the two referees for taking the time to review our paper and for their helpful comments. For the details, please look into the paper with keeping track of changes.*

Referee #2

General:

This paper describes the variations and sources of CO, $C_2H_2$, $C_2H_6$, $H_2CO$ and HCN columns observed with ground-based FTIR at Xianghe in North China along with the retrieval methods. The paper is well described, and it should be published after some minor revisions.

Comments and questions:

p3, l7 Please add the altitude of the Xianghe site.

Done

p3, l25-28. If you want to describe only InSb measurements, the number of the optical filter should be 5. I think it is better to describe MCT measurements even you don't use the spectra in this paper.

Thanks for the suggestion, but currently we only operate the InSb and InGaAs measurements, and the MCT measurements are not recorded at Xianghe.

p5, l2 Please add the version of SFIT4.

Done

l3-4 Full name of NDACC-IRWG was already described in section 1.

Corrected

l29 What kind of a priori profiles were used for $H_2O$, HDO, $H_2^{17}O$ and $H_2^{18}O$?

Added. They are derived from the NCEP reanalysis data.

Table 3 Wavenumbers for the retrieval windows of $H_2CO$ and HCN are incorrect.

Thanks, it is corrected now.

Figure 3 Don't you show the observed spectra?

We prefer to keep it more readable like now.

p11, l10 $10^{19}$ should be $10^{18}$.

Corrected.

Figure 5 The yellow shaded area is hard to see because it is almost covered by gray dots.The box plots become narrow at the median value (Only the box plot for XCO in Figure A1 is box shape). As for outliers, did you exclude them in the correlation analysis? I think you used them because they have strong information of the local anthropogenic emissions. So, the word 'outliers' is misleading.

The Figure 5 is updated. The 'outliers' are used in the correlation analysis.

We agree with the referee that the 'outliers' is not appropriate. We changed it to 'extremely high/low values'.

p13 l6 42% maybe wrong. 62%?

Corrected

l27-33 Is this paragraph worth writing in this paper?

The referee #1 also suggest to remove it. We have removed this paragraph now.

p14 l16 less --> smaller?

Done

p17, l4 southeast --> southwest?

Corrected

l17 south-east --> southwest?

Corrected

p18 l3trace --> tracer

Corrected

l5-6 Schroeder et al.(2014) --> (Schroeder et al., 2014)

Corrected

l9 Cao et al. (2014) --> (Cao et al., 2014)

Corrected

p20 l1

 are summed up before each airmass releasing day --> before each airmass releasing day are summed up

Done.

3.5 p21 l20 Emission estimation for $C_2H_2$ and $C_2H_6$' maybe better.

Done.

p22 Figure 13 It is better to put year in figure (b) and (c).

Done.

l6-7 Is this mean that you used the slopes shown in Figure 6? Is it better than using data in categories B and C (or only C)?

Yes, the slopes used here are derived using all categories. In fact, the slopes using all categories and only categories B and C are close to each other.

p23 Can you add some more discussion for the difference of the $C_2H_2$ and $C_2H_6$ emissions between the inventories and your estimations? The emissions of the two species are nearly the same in two inventories while fossil fuel/biofuels source for $C_2H_6$ is twice larger than that of $C_2H_2$ in Table 1. The source values in Table 1 seems to be consistent with your estimations.

Thanks for the suggestion. The C2H2 and C2H6 emissions are almost the same from both inventories. The reason is probably because that the emission factors of C2H2 and C2H6 inventories are both based on the EMEP/EEA guidebook (Huang et al., 2017; Kurokawa et al., 2020). However, the uncertainty of the emissions of the NMVOCs is pretty large. The emissions of C2H2 and C2H6 derived from our FTIR measurements at Xianghe indicates that the emission factors used for C2H2 and C2H6 need to be improved, and the FTIR measurements suggest that the C2H6 emission is about 2.2 times larger than the C2H2 emissions.

p23, l2010[19] should be 10[18].

Corrected.

 0.92 --> 0.91?

Corrected.

p24 l5-7 Is this sentence important conclusion? I think it is enough to put this sentence in section 3.1.

Done.

l14-15 come from local anthropogenic emissions' is explained in the next sentence.

Corrected.

Appendix A

p26 l3  'Since' should be removed.

Done

Figure A1

 Dashed line is hard to see.

Added the explanation: 'the dashed line is almost overlapped with the red solid line'

---

## Author Response (AR2)

Frank's comment:

thanks so much for the revised manuscript. This all looks fine, apart from one minor explanation you added as result of the communication with one of the referees (page 5, line 29): "the average of the WACCM v7 monthly means between 1980 and 2040 are applied for the a priori profiles of the target species and other interfering species except for $H_2O$, HDO, $H_2^{17}O$ and $H_2^{18}O$ (NCEP reanalysis)." To my knowledge, the NCEP reanalysis only provides $H_2O$ profiles, but no information about the other $H_2O$ isotopologues. Could you please be more specific on this point? (I guess you create the profiles of the isotopologues using some climatological abundance ratio profiles wrt the main isotopologue?)

Dear Frank,
Thanks for your comment. You are right that NCEP reanalysis only provides $H_2O$ profiles but no information about other $H_2O$ isotopes. Here, we use the NCEP reanalysis to set $H_2O$ profiles as well as $H_2O$ isotopes, assuming that the $H_2O$ isotopes have the same variations as the main isotope ($H_2O$). Since the line intensities are scaled by the terrestrial isotopic fraction so that the same vmrs of $H_2O$ and its isotopes will retrieve the correct amount from all isotopic lines of a particular. Of course, if we want to report the retrieved $H_2O$ isotopes, we need to multiply the terrestrial isotopic fraction to the retrieved VMR.

Yours,
Minqiang and coauthors